# Single-Cell Proteomic Analysis Dissects the Complexity of Tumor Microenvironment in Muscle Invasive Bladder Cancer

**DOI:** 10.3390/cancers13215440

**Published:** 2021-10-29

**Authors:** Chao Feng, Xi Wang, Yuting Tao, Yuanliang Xie, Zhiyong Lai, Zhijian Li, Jiaxin Hu, Shaomei Tang, Lixin Pan, Liangyu He, Qiuyan Wang, Tianyu Li, Zengnan Mo

**Affiliations:** 1Center for Genomic and Personalized Medicine, Guangxi Medical University, Nanning 530021, China; fengchaogx@163.com (C.F.); wangxi@stu.gxmu.edu.cn (X.W.); tytxzxh@126.com (Y.T.); xieyuanliang123@163.com (Y.X.); laizhy92847684@live.com (Z.L.); zhijianli1996@163.com (Z.L.); huga102@163.com (J.H.); 18169703093@163.com (S.T.); pankeke5@163.com (L.P.); heliangyu123456@163.com (L.H.); zengnanmo@hotmail.com (Z.M.); 2Guangxi Key Laboratory for Genomic and Personalized Medicine, Guangxi Collaborative Innovation Center for Genomic and Personalized Medicine, Nanning 530021, China; 3Department of Biochemistry and Molecular Biology, School of Basic Medical Sciences, Guangxi Medical University, Nanning 530021, China; 4Key Laboratory of Longevity and Aging-Related Diseases of Chinese Ministry of Education, Center for Translational Medicine, Guangxi Medical University, Nanning 530021, China; 5Department of Urology, Affiliated Tumor Hospital of Guangxi Medical University, Nanning 530021, China; 6Institute of Urology and Nephrology, The First Affiliated Hospital of Guangxi Medical University, Nanning 530021, China; 7Departments of Urology, The First Affiliated Hospital of Guangxi Medical University, Nanning 530021, China

**Keywords:** muscle invasive bladder cancer, tumor microenvironment, cancer stem cell, mass cytometry, imaging mass cytometry

## Abstract

**Simple Summary:**

The tumor microenvironment (TME) is considered to play a key role in the development of many types of tumors. Muscle invasive bladder cancer (MIBC), which is well known for its heterogeneity, has a highly complex TME. Herein, we integrated mass cytometry and imaging mass cytometry to systematically investigate the complexity of the MIBC TME. Our investigation revealed tumor and immune cells with diverse phenotypes. We identified a specific cancer stem-like cell cluster (ALDH^+^PD-L1^+^ER-β^−^), which is associated with poor prognosis and highlighted the importance of the spatial distribution patterns of MIBC TME components. The present study comprehensively elucidated the complexity of the MIBC TME and provides potentially valuable information for future research.

**Abstract:**

Muscle invasive bladder cancer (MIBC) is a malignancy with considerable heterogeneity. The MIBC tumor microenvironment (TME) is highly complex, comprising diverse phenotypes and spatial architectures. The complexity of the MIBC TME must be characterized to provide potential targets for precision therapy. Herein, an integrated combination of mass cytometry and imaging mass cytometry was used to analyze tumor cells, immune cells, and TME spatial characteristics of 44 MIBC patients. We detected tumor and immune cell clusters with abnormal phenotypes. In particular, we identified a previously overlooked cancer stem-like cell cluster (ALDH^+^PD-L1^+^ER-β^−^) that was strongly associated with poor prognosis. We elucidated the different spatial architectures of immune cells (excluded, infiltrated, and deserted) and tumor-associated collagens (curved, stretched, directionally distributed, and chaotic) in the MIBC TME. The present study is the first to provide in-depth insight into the complexity of the MIBC TME at the single-cell level. Our results will improve the general understanding of the heterogeneous characteristics of MIBC, potentially facilitating patient stratification and personalized therapy.

## 1. Introduction

Muscle invasive bladder cancer (MIBC) is a highly aggressive malignancy, and the ninth leading cause of cancer-related deaths worldwide. It is associated with a 40–60% survival rate at 5 years [1,2]. MIBC is characterized by heterogeneity, and high rates of recurrence and mortality [3,4]. Approximately 20–25% bladder cancer patients are diagnosed with MIBC during their initial diagnoses [5]. Despite an improvement in MIBC survival rates with cisplatin-based neoadjuvant chemotherapy (NAC) followed by radical cystectomy, precise treatment options for MIBC patients are urgently needed [6]. 

The tumor microenvironment (TME) comprises multiple types of cells, including tumor, immune and stromal cells. The heterogeneous phenotypes, quantities and locations of TME components affect the complexity of TME [7]. Complex TMEs play pivotal roles in tumor progression, metastasis, recurrence, and significantly affect therapeutic responses [8]. In the TME, tumor cells are stimulated by metabolites, stromal cells, and signaling molecules, owing to which they exhibit remarkable plasticity [9]. Plasticity enables tumor cells to constantly convert between differentiated states and cancer stem cell (CSC) states [10]. This allows them to adjust to ever-changing microenvironments, evade immune attack, invade and disseminate [11]. Furthermore, an immunosuppressive TME promotes immune evasion, and is associated with a poor prognosis. It recruits abundant exhausted T cells and regulatory T cells (Tregs). Both are, in fact, the main targets of immune checkpoint inhibition therapies that have recently revolutionized cancer therapy [12,13]. Exhausted T cells are characterized by dysfunction, and persistently express multiple inhibitory receptors (such as PD-1, CTLA-4, Tim-3, and LAG3) [14]. Tregs play immunosuppressive functions by secreting immunosuppressive cytokines [15]. Analyses of various cancers have shown that T cell exhaustion and Tregs drive tumor progression [14,16]. 

With the rapid development of single-cell technologies, studies have comprehensively explored the heterogeneity of MIBC [17,18]. Many clusters of tumor, immune, and stromal cells with important functions have been identified. Using single-cell RNA sequencing (scRNA-seq), studies have elucidated that tumor cells involved in bladder cancer lose the ability to express MHC-II molecules, and that inflammatory cancer-associated fibroblasts accelerate tumor progression [19]. However, scRNA-seq cannot illustrate the spatial distribution characteristics and interactions between neighboring cells in different types of cell clusters. In contrast to scRNA-seq, an integrated combination of mass cytometry (CyTOF) and imaging mass cytometry (IMC) can be used to systematically investigate the diverse phenotypes of the cells in the TME at the single-cell level. They can also be used to explore the spatial location characteristics of such cells. Therefore, the two techniques constitute a practical strategy for TME research [20,21].

Herein, we integrated CyTOF and IMC to explore the complexity of the MIBC TME. CyTOF analysis revealed the abnormal phenotypes of tumor cells and immune cells, and IMC identified the major spatial phenotypes of the MIBC TME. This single-cell proteomic analysis demonstrates the complexity of the MIBC TME and provides a foundation for precise TME-targeted MIBC therapy.

## 2. Results

### 2.1. TME Landscape in MIBC

To map the TME landscape of MIBC, 79 samples from 44 patients (Appendix A), including 44 cancer (CA) tissues and 35 para-carcinoma (CP) tissues, were evaluated by CyTOF. t-SNE algorithms were performed to reduce the high dimensional data and generated two-dimensional map of data. The analysis of single cells in the MIBC TME revealed various unique protein expression patterns (Figure 1A). In order to detect the specific clusters in the TME, phenograph algorithm was utilized to identify 21 clusters with distinct phenotypes (Figure 1B). The analysis of clusters from CA and CP tissues showed that the frequencies of many clusters in CA and CP tissues differed, indicative that the constitution of the TME in CA and CP tissues varied (Figure 1C). In the 21 specific clusters, 6 immune cell clusters represented as CD326^−/low^CD45^+^, and 5 tumor epithelial cell clusters expressed CD326^+^CD45^−/low^, while the clusters showed that CD326^−/low^ CD45^−/low^ were defined as “other cells” which maybe dedifferentiated tumor cells, fibroblast, endothelial cells or other cell types. Interestingly, we detected that 7 specific clusters positively expressed CD326 and CD45 which possessed the characteristics of immune cell and tumor epithelial cell simultaneously. These 21 specific clusters exhibited highly heterogeneity among individual samples (Figure 1E). In the MIBC TME, tumor cell clusters exhibited diverse protein expression patterns. Cluster 1 highly expressed ICAM-1 (CD54) and vimentin, positively expressed cancer stem cell markers CD90 and LGR5 [22,23]; Cluster 2 characteristics of high levels of CD47, Notch 2, vimentin, CD90, LGR5, Sox2; Cluster 3 exhibiting high levels of PD-L1, androgen receptor (AR), bladder cancer stem cell markers CD133, ALDH [24,25]; Cluster 6 was characterized by high levels of PD- L1, ALDH, and lacked expression of ER-β. All these 4 clusters were characterized by cancer stem-like cell phenotypes; Cluster 10 and 18 exhibited high levels of vimentin, ER-β and ICAM-1, positive expression of PD-L1 (Figure 1D). To further explore the characteristics of the specific clusters, we stratified patients based on the frequency of each cluster, into high- and low-abundance groups. Notably, the SPADE analysis showed that the tumor epithelial cells of cluster 10 high-abundance group patients highly expressed c-Myc, Ki67, and vimentin, when compared with low-abundance group patients, indicating that the high-abundance group patients may exhibit epithelial-mesenchymal transition (EMT) state and a high proliferation rate. Cluster 18 exhibited the same patterns as cluster 10 (Figure 2A,B).

To reveal the relationships of clusters in MIBC ecosystem, spearman rank correlation analysis was performed. Many relationships among the clusters in the MIBC TME were revealed (Figure 2C). Cancer stem-like cell cluster 3, CD326^+^CD45^+^ tumor epithelial cell cluster 10 and 18 existed robust relationships with each other (Figure 2D), indicating that these clusters potentially played similar roles in the TME, while multiple strong relationships were found between tumor cells clusters and immune cells clusters (Figure 2C). The above results suggested that the interactions of clusters contributed to the complexity of the MIBC TME.

### 2.2. A Specific Cancer Stem-Like Cell Cluster Associates with Poor Prognosis

Interestingly, we identified a previously unappreciated cancer stem-like cell cluster ALDH^+^PD-L1^+^ER-β^−^ (Cluster 6) which was strongly associated with poor clinical outcome (Figure 1D and Figure 3A). When patients were divided into high- and low-abundance groups according to the frequency of cluster 6, we found that cluster 6 high-abundance group patients were associated with advanced stage and age (Figure 3B,C). Meanwhile, tumor dedifferentiation occurred in 63.6 percent of cluster 6 high-abundance group patients, which lost expression of epithelial cell markers Pan-CK and E-cadherin, while only 27.8 percent of low-abundance group patients appeared in dedifferentiation states (Figure 3F,G). Furthermore, spatial analysis from IMC showed that cluster 6 was located in the stromal regions of the tumor invasive front (Figure 3D,E), representing a phenotype of the tumor budding cell [26]. 

To further understand the molecular characteristics of the high- and low-abundance group’s patients in cluster 6, we next analyzed the DEGs between these two groups and found that many DEGs were enriched in neuronal cell body, positive regulation of neurogenesis, neurotransmitter transport, neuro migration, and other neuro-related pathways (Figure 3H). Transcription factor enrichment analysis of the DEGs showed that in 7 of the top 10 potential transcription factors of the DEGs: SCRT1, DPF1, ZnF488, PIN1, CUX2 previously promoted neural differentiation and neuroendocrine cancer progression, while EMX1 and MYRF were reported to promote the development of nervous system [27,28,29,30,31,32,33], implying that cluster 6 might be regulated by neural related genes (Figure 3I). Furthermore, we observed that glycosphingolipid synthase B4GALNT1 was significantly upregulated in cluster 6 high-abundance group (Figure 4A). It was reported previously to affect neuro-tumor progression and cancer stem cell characteristics by regulating glycosphingolipids, which are enriched in exosomes [34,35,36]. In TCGA bladder urothelial carcinoma (BLCA) cohort, high levels of B4GALNT1 were significantly related to clinical stages and poor prognosis (Figure 4B,C). In our cohort, the genes significantly related to B4GALNT1 expression were enriched in neurotransmitter transport, positive regulation of neuron differentiation, and many other neuro-related pathways (Appendix A). Immunohistochemical (IHC) staining and hematoxylin-eosin (HE) staining results showed that B4GALNT1 was also located in stromal region (Appendix A), implying a robust relationship between B4GALNT1 and cluster 6.

Then, we downloaded the published scRNA-seq data of 8 bladder cancer tissues to further explore the underlying mechanisms of B4GALNT1 [19]. According to the expression levels of B4GALNT1, we divided 8 tissues into high- and low-expression groups. 32,511 single cells from 8 tissues were clustered into 25 clusters (Appendix A). EPCAM and COL1A1 were used to identify tumor epithelial cells and fibroblasts, as described in a previous study [37] (Appendix A). Obviously, B4GALNT1 was enriched in fibroblast clusters (Appendix A), consisting of our IHC and HE results. DEGs enrichment analysis showed that the major upregulated DEGs between fibroblasts in high- and low-expression group of B4GALNT1 were significantly enriched in extracellular space, extracellular region, ECM and the extracellular exosome (Figure 4D). Furthermore, a GO analysis of upregulated DEGs between tumor epithelial cells in high- and low-expression group showed that extracellular exosome was the top-ranked signaling pathway; also, extracellular space and ECM were in the top 20 signaling pathways (Figure 4E). In addition, several heat shock proteins: HSPH1, HSPD1, HSPA6, HSPA1B, HSPA1A, HSP90AA1 were upregulated in the fibroblasts of high-expression group (Figure 4F). Indeed, studies reported that heat shock proteins are enriched in exosomes, which can promote cell-cell crosstalk in TME [38,39]. Importantly, neighborhood analysis from IMC can precisely illustrate the cell-cell interactions of specific cell clusters in TME according to the location information of single cells [40]. Here, neighborhood analysis showed that the interaction rates of the clusters in the TME of B4GALNT1 high-expression patients were significantly higher than that of low-expression patients (Figure 4G,H), and the spatial distribution of the clusters in B4GALNT1 high-expression patients was more chaotic than that in low-expression patients, indicative of more cell-cell crosstalk occurred in the TME of B4GALNT1 high expression patients (Figure 4I,J). Taken together, all these data may suggest that B4GALNT1 promoted crosstalk among cell clusters by activating extracellular exosomes, thus facilitating MIBC progression.

### 2.3. The Heterogeneous Phenotypes of Immune Cells in the MIBC TME

Immunotherapy recently revolutionized cancer therapy, which is influenced by tumor immune microenvironment [12,13]. To map the immune microenvironment landscape of MIBC, we design a panel which contained 34 antibodies to analyze the protein expression patterns of immune cells in MIBC ecosystem. To acquire the protein expression of immune cells, we gated the immune cells in cytobank platform with immune cells specific marker CD45. Then, t-SNE and phenograph algorithms were used to deal with high dimensional data and identify 19 immune cell clusters with distinct phenotypes in MIBC ecosystem (Figure 5A), including 6 CD4^+^ T cell clusters, 5 CD8^+^ T cell clusters, 2 B cell clusters, 3 DC cell clusters, 2 macrophage clusters, and a NK cell cluster (Figure 5C). The frequencies of clusters in CA tissues and CP tissues were greatly different, indicating that the immune microenvironment of CA tissues and CP tissues were distinctive (Figure 5A,B). We further analyzed the frequencies of clusters in individual cancer samples, found that a distinct immune cell cluster with an absolute advantage in number existed in each sample, and suggested that the immune cells were heterogeneous among individual samples (Figure 5D).

T cells play a dual role of anti-cancer and pro-cancer activities in the immune microenvironment [41,42]. In MIBC ecosystem, 11 T cells clusters with diverse expression patterns of immune checkpoint, co-inhibitory receptor and activation markers, as shown in Figure 5C, represented exhaust or immunosuppressive phenotypes [43]. Cluster 16 characterized by high levels of PD-1 (CD279), TIM-3, GranzymeB, CD27, positively expressed in CD95, HLA-DR, CTLA-4, ICOS (CD278), was an exhausted CD8^+^ T cell cluster. Cluster 10, cluster 12, and cluster 13 negatively expressed of PD-1, CTLA-4 and HLA-DR, while cluster 17 had the same markers expression pattern as cluster 16, but low expression levels of PD-1, ICOS, CTLA-4, HLA-DR, and CD95. Cluster 19 high expressed CD3, CD4 and Foxp3, was identified as a Treg cell cluster, which positively expressed PD-1, TIM-3, HLA-DR, high expressed CTLA-4, CD28, CD27, ICOS. The cluster 11, cluster 14, cluster 15, and cluster 18 played a similar markers expression pattern as cluster 19, but with low expression levels (Figure 5C,E). When dividing patients into high- and low-abundance group, according to the frequencies of each cluster, we found that the frequencies of infiltrating T cells, including CD4^+^ T cells, CD8^+^ T cells, Tregs and PD-1^+^ T cells in cluster 16 high-abundance group, were higher than in the low-abundance group (Figure 5F). Meanwhile, the frequencies of PD-1^+^, TIM-3^+^ and CTLA-4^+^ immune cells of cluster 16 high-abundance group were also significantly higher than the low-abundance group (Figure 5G and Appendix A). Furthermore, the immune checkpoint, co-inhibitory receptor and activation markers expression levels of CD8^+^ T cells, Tregs and macrophages in cluster 16 high-abundance group were prominently higher than the low-abundance group (Appendix A). Cluster 19 played similar patterns with cluster 16 (Figure 5F,G). This suggests that cluster 16 and cluster 19 may be associated with the immunosuppressive microenvironment of MIBC. 

### 2.4. The Spatial Resolution-Based Phenotypes of the MIBC TME

In a personalized precision medicine era, the subgrouping of patients based on molecular characteristics is emerging as a critical factor for prognosis and treatment of cancers [44]. Recently, a consensus molecular classification (ConMC) summarized the merits of the published classifications and converged MIBC on six subtypes, based on 1750 MIBC transcriptomic profiles from 18 published datasets [45]. The ConMC provides a new framework for MIBC research. In this study, we grouped 38 samples into six molecular subtypes based on the ConMC (Appendix A), and analyzed the characteristics of the TME among the six molecular subtypes at the single-cell level. However, the TME components of the six molecular subtypes were no significant differences, indicative of the great heterogeneity of the MIBC TME, and the ConMC did not lay a foundation for the MIBC TME research (Appendix A). 

Next, to further dissect the complexity of the MIBC TME, we performed IMC to explore the spatial resolution-based phenotypes of the MIBC TME. Pan-CK (epithelium), Collagen I (extracellular matrix), and CD45 (immune cell) were visualized to identify morphological features of the MIBC TME. In the TME, we observed several major spatial phenotypes of tumor cells, immune cells, and collagen signatures. In tumor regions, two types of the TMEs were occurred; type I region showed a strong expression of the epithelial cell markers Pan-CK and E-cadherin (Figure 6A). Meanwhile, type II region lost the expression of Pan-CK and E-cadherin, indicative of the occurrence of tumor dedifferentiation (Figure 6B). It always occurred in poor prognosis samples. Immune cells in the MIBC TME also exhibited three major spatial distribution patterns. The immune cells enriched in type I region were excluded by tumor cells, which cloud not infiltrated tumor nests (Figure 6C). Type II region recruited the immune cells infiltrating tumor nests (Figure 6D). Type III region showed few immune cells (Figure 6E). Similarly, four major spatial distribution patterns of collagen signatures occurred in the MIBC TME. The curved collagen fibers wrapping around tumor nests were enriched in type I region (Figure 6F). The collagen fibers enriched in type II region stretched and aligned more parallel to the tumor boundary (Figure 6G). Type III region contained the directionally distributed collagen fibers (Figure 6H). Meanwhile, type IV region displayed chaotically aligned collagen fibers without clear tumor boundaries (Figure 6I). All these results suggested that the MIBC TME was highly complex; this is why the treatment responses of MIBC patients were extremely different.

## 3. Discussion

MIBC is a highly heterogeneous disease. As a key driver of tumor progression, the TME endows tumor cells with considerable plasticity and the ability to alter their phenotype to adapt to the changing circumstances. In the present study, we revealed the abnormal phenotypes of tumor epithelial cells and immune cells in the MIBC TME, and highlighted the complexity of the TME. Previous investigations have used bulk RNA sequencing and scRNA-seq to gain insight into bladder cancer TME [8,19,46]. Although these investigations have produced some important discoveries, the results are limited. Therefore, we integrated CyTOF and IMC to explore the complexity of the MIBC TME. 

In the present study, CyTOF was performed to assess the phenotypic diversity of the MIBC TME at the single-cell level. The single-cells proteomics analysis expanded our perspective on the abnormal phenotypes of tumor epithelial cells and immune cells. Using CyTOF, we demonstrated that some specific tumor epithelial cells clusters are characterized by diverse expression patterns of markers that are specific for metastasis, immune checkpoint, cancer stem cell, and cancer relative pathways. We identified a previously overlooked cancer stem-like cell cluster ALDH^+^PD-L1^+^ER-β^−^ (cluster 6). This cluster was located in the stromal region of the tumor invasive front, and represents a phenotype of tumor budding cells. Many studies have reported that tumor budding cells are typically located at the tumor invasion front, reflecting the EMT [47]. Tumor budding cells express high levels of the cancer stem cell markers LGR5, ALDH1A, CD44, and always escape immune surveillance [26]. Recent studies have suggested that tumor budding cells are characterized by malignant and active invasion, and are independent prognostic indicators of many tumor types [48,49,50]. Therefore, cluster 6 may be invasive and malignant. When the patients were stratified into high- and low-abundance groups based on the frequency of cluster 6, the high-abundance patients had poor prognoses. Furthermore, many transcriptome DEGs between high- and low-abundance groups highly enriched in neuro-related pathways. It is possible that cluster 6 is regulated by neural-related genes. Neuro-signaling can promote cancer cell growth and metastasis by regulating the TME [51]. There is also evidence that metastatic tumors acquire a remarkable property associated with neurons and that tumor cells with neuronal features are always endowed with plasticity, which affects the aggressiveness of the tumor [52,53]. Furthermore, glycosphingolipid synthase B4GALNT1, which has previously been associated with neuro-tumor progression and cancer stem cell characteristics by affecting glycosphingolipids, was upregulated in the high-abundance group [34,35]. The IMC and IHC analysis showed that both cluster 6 and B4GALNT1 were located in the stromal region. Therefore, the results described above indicate that cluster 6 is related to neural cells and is potentially regulated by B4GALNT1. Bladder cancer with neural features is rare, but is a lethal malignancy with a high rate of metastasis and a median overall survival expectation of 9–20 months [54,55]. Perhaps cluster 6 contributes to the high malignancy and plays a key role in MIBC progression. However, when we overexpressed B4GALNT1 in bladder cancer cell lines, there were no observable phenotypic changes, implying that B4GALNT1 might facilitate MIBC progression by cooperating with stromal cells. Previous studies reported that cancer-associated fibroblasts promote cisplatin resistance, induce EMT, and associate with poor prognosis in bladder cancer [19,56,57]. Therefore, to further confirm the mechanism by which B4GALNT1 acts in MIBC, we analyzed the published scRNA-seq data of eight bladder cancer tissues samples, and found that the major upregulated DEGs of fibroblasts in the high- and low-B4GALNT1 expression groups were significantly enriched in the extracellular exosome and the ECM-related signaling pathways. The DEGs in tumor epithelial cells exhibited a similar pattern. The fibroblasts in the high-B4GALNT1expression group expressed high levels of heat shock proteins, which have been reported to be enriched in exosome and to promote exosome release [58]. This suggests that ECM remodeling and activation of exosome mechanism possibly occurred in the high-B4GALNT1 expression group. Extracellular exosome can improve tumor innervation and intercellular crosstalk to facilitate tumor dissemination [59]. In addition, neighborhood analysis from IMC illustrated that the high-B4GALNT1 expression patients had more chaotic TMEs than the low-B4GALNT1 expression patients, which implied strong crosstalk among the specific clusters. Therefore, we speculate that B4GALNT1 induces ECM remodeling, and activates extracellular exosomes to promote cell–cell crosstalk and accelerate MIBC progression.

With the successful application of immunotherapy, tumor immune microenvironment has become the focus of considerable attention. In the present study, CyTOF revealed that the T cell clusters expressed PD-1 to different degrees, and the frequency of PD-1^+^ T cells varied among individual samples. Furthermore, the T cell clusters in the MIBC TME co-expressed activation markers and co-inhibitory receptors, which were characterized by exhausted phenotypes, as reported previously [60]. 

Subgrouping of patients based on molecular characteristics has considerably affected the selection of therapeutic regimens and the forecasting of therapeutic responses and prognoses [44]. The establishment of tumor molecular subtypes has advanced our understanding of tumor genotypes and phenotypes, presenting a potential application in therapeutics and prognosis estimation [61,62]. At present, the ConMC divides MIBC into six molecular subtypes and systematically exposes specific differentiation patterns, mutation genes, histology, and overall survival rates in the six molecular subtypes [45]. The study provides a foundation for the study of MIBC heterogeneity. Herein, based on the ConMC, we divided 38 samples into six molecular subtypes, but did not observe any significant differences in the TME components of the MIBC subtypes. Suggesting that the MIBC TME has great heterogeneity, the ConMC may not be suitable for studying the remarkable heterogeneity of the MIBC TME.

Cancer metastasis and progression originate from abnormal interactions between tumor cells and the TME [63]. Both the quantity and the spatial distribution of TME compositions contribute to the complexity of the TME and impact cancer progression [64]. We observed several types of TME in situ in MIBC. In the tumor regions, we detected two types of TMEs: the first type was characterized by a high rate of expression of the epithelial cell markers Pan-CK and E-cadherin, and the second type was characterized by a lack of expression, representing a dedifferentiated phenotype, which always occurs in malignant and invasive tumors [65]. The immune cells in the MIBC TME also exhibited three major spatial distribution patterns. The type I region always recruits immune cells, and is excluded by tumor cells, which may impair anti-tumor immune responses. Immune cells enriched in the type II region can infiltrate tumor nests, where they may fully exert their anti-tumor effect. In contrast, the type III region had an immune desert phenotype. IMC also revealed four major spatial distribution patterns of tumor-associated collagen signatures in the TME. In the type I region, the curved collagen fibers wrapped around tumor nests were not conducive to the metastasis of tumor cells. The type II region was enriched with stretched collagen fibers aligned parallel to the tumor boundary, which may be beneficial to tumor growth and metastasis. The collagen fibers enriched in the type III region were directionally distributed, which may contribute to the unidirectional migration of tumor cells. In the type IV region, chaotically aligned collagen fibers were enriched; moreover, as these collagen fibers may contribute to the multidirectional metastasis of tumor cells, the tumor was indicated to be malignant [66,67]. Collectively, the results from the present study suggest that the MIBC TME is highly complex and comprises diverse phenotypes. The spatial resolution-based phenotypes of the TME, which contribute to the complexity of the MIBC TME, deserve more attention. 

However, it should be noted that the present study has some limitations. First, CyTOF and IMC are dependent on high-quality antibodies. Owing to the limited availability of commercial antibodies, we could only analyze the characteristics of the TME based on existing antibody tools. Second, the present study only described the complexity of the MIBC TME; the functions and genetic profiles of the clusters identified herein require further research. In particular, the functions and underlying mechanisms of the ALDH^+^PD-L1^+^ER-β^−^ cancer stem-like cell cluster require urgent investigation. Third, all the phenotypes of clusters identified herein were based on a cohort of 44 patients with MIBC. Larger and independent cohorts should be analyzed to reveal additional valuable clinical phenotypes. 

In summary, we integrated CyTOF and IMC to understand the complexity of the MIBC TME at the single-cell level. The results emphasized various spatial phenotypes of the MIBC TMEs. Therefore, the present study has expanded the background knowledge required for precision MIBC therapy.

## 4. Materials and Methods

### 4.1. Patients and Samples

Seventy-nine tissue samples were collected from 44 MIBC patients who underwent radical cystectomy in the First Affiliated Hospital of Guangxi Medical University during February 2018 and October 2019. All participants had not received any tumor therapy before enrolment, and the diagnosis of MIBC was confirmed by two experienced pathologists. Paraffin histopathological sections were obtained from the pathology department of the First Affiliated Hospital of Guangxi Medical University. Written informed consent was provided by all participants. This study was approved by the ethics committee of the First Affiliated Hospital of Guangxi Medical University (2019(KY-E-147)).

### 4.2. Cell Isolation

The fresh bladder cancer tissues were transferred from the operating room to the laboratory in cold HBSS (311-512-CL; WISENT, Saint-Jean-Baptiste, Quebec, Canada) with 1% penicillin-streptomycin (15240062; Gibco, Shanghai, China) within 20 min of removal. Bladder tissues were washed with cold Dulbecco’s phosphate-buffered saline (DPBS, without Mg^2+^ and Ca^2+^, 311-425-CL; WISENT), minced into small fragments with surgical scissors, and centrifuged at 300× *g* for 5 min at 4 °C. After discarding the supernatant, the tissue fragments were transferred to a tissue detach tube (Miltenyi, Bergisch Gladbach, Germany) and perfused with 8 mL of collagenase type I (1.5 mg/mL; 17100017; Gibco) supplemented with DNase I (0.2 mg/mL; 10104159001; Roche, Basel, Switzerland). Next, the tissue was dissociated with GentleMACS tissue dissociator (Miltenyi) according to manufacturer’s instruction at 37 °C for 20 min. After centrifugation (5 min at 300× *g*, 4 °C), residual tissue fragments were dissociated again. Enzymatic dissociation was terminated with 10 mL DMEM (319-006-CL; WISENT) supplemented with 10% FBS (10099141; Gibco). Next, the cell suspension was filtered through a 70 μm cell strainer and washed with red blood cell (RBC) lysis buffer (Solarbio, Beijing, China) to remove red blood cells. Finally, the single cell suspension was washed with DPBS again, and cells were frozen in FBS, complemented with 10% DMSO.

### 4.3. RNA-Sequencing

Bulk RNA sequencing was performed as previously described [68]. Briefly, total RNA of 38 MIBC tissues was extracted by Trizol (Life Technologies Corporation, Carlsbad, CA, USA). Total RNA concentration was estimated with a Nanodrop spectrophotometer (Thermo Fisher Scientific, Waltham, MA, USA). RNA sequencing was performed using the Illumina NextSeq 500 platform (Illumina, San Diego, CA, USA). Samples were successfully classified into six molecular subtypes by the ConMC based on transcriptomic data [45]. Data from the cancer genome atlas (TCGA) were analyzed by a bioinformatics tool GEPIA (http://gepia.cancer-pku.cn/detail.php (accessed on 19 August 2020) with the default setting [69]. Transcription factor enrichment analysis was performed with the bioinformatics tool ChEA3 (https://maayanlab.cloud/chea3/#top (accessed on 4 March 2021)) with the default setting [70]. Single cell RNA sequencing data of 8 bladder cancer tissues were mined from a recent publication [19]. The quality control of raw data was according to published pipelines, and data processing was performed as previously described [71]. Briefly, the integrated data matrix was subjected to dimensionality reduction by principal component analysis (PCA). 25 clusters were identified with the FindClusters function in Seurat, and they were visualized with 2D t-SNE plots. EPCAM and COL1A1 were used to identify tumor epithelial cells and fibroblasts. Differentially expressed genes (DEGs) of tumor epithelial cells or fibroblasts between B4GALNT1 high- and low-expression groups were identified by Seurat FindMarkers function. Gene ontology (GO) analysis was performed with clusterProfiler.

### 4.4. Antibodies and Antibody Labeling

Antibodies used for CyTOF in this study were listed in Appendix A. Some preconjugated antibodies were purchased from supplier (Fluidigm, San Francisco, CA, USA) directly, while others were labeled with specified metal tag using the MaxPAR antibody conjugation kit (Fluidigm) according to the manufacturer’s instruction. 

### 4.5. Mass Cytometry

Cryopreserved cells were recovered and single cells from 4 samples were barcoded with barcoding reagents simultaneously, as previously described [72]. Surface staining for CyTOF was performed as previously described, with slight modifications [73]. Briefly, Cisplatin (Fluidigm) staining was used to identify dead cells, and then cells were stained with antibody cocktails. Cells were then incubated with Cell-ID intercalator-Ir (Fluidigm) overnight at 4 °C to identify cellular events and data acquired with CyTOF2^TM^ mass cytometer (Fluidigm). EQ™ four element calibration beads (201078, Fluidigm) were used to normalize signals according to the manufacturer’s instructions.

FCS files generated by CyTOF2^TM^ mass cytometer were deconvoluted (debarcoded) and uploaded to Cytobank platform (www.cytobank.org (accessed on 11 December 2019)). To clear data, events recorded were gated according to DNA sign intensity and cell length. Populations of interest were gated manually, and the events of population were exported as FCS files and .csv files. The signal values of channels acquired by CyTOF were converted using arcsinh transformed with a cofactor of 5, and the characteristic clusters were identified by performing phenograph on 79 samples, which randomly extracted 5000 cells per sample to analyze using cytofkit R package [74]; t-distributed stochastic neighbor embedding (t-SNE) was used to convert high-dimensional data into two dimension with default setting: perplexity value 30, iteration value 1000, using the t-SNE R package; and phenotype heatmaps of clusters were drawn using pheatmap R package. Spearman rank correlation analysis was displayed in R using the Corrplot R package, with the frequency of clusters as input [75].

### 4.6. Imaging Mass Cytometry

Paraffin slices (thickness of 5 µm) were incubated at 65 °C for 2 h in an oven. Slices were then incubated with xylene to further deparaffinized and carried through sequential rehydration from absolute ethanol to 75% ethanol before washing with double distilled water. Antigen retrieval was performed in a 50 mL EP tube, which contained 40 mL Tris-EDTA buffer pH 9.0 (FC16FA0005, Sangon Biotech (Shanghai) Co., Ltd., China) for 30 min at 96 °C. The 50 mL EP tubes were slowly cooled at room temperature for 20 min. After washing with PBS, slices were blocked with 3% BSA in DPBS for 45 min at room temperature, and then incubated with a metal-conjugated antibody cocktail (diluted with DPBS solution contained 0.5% BSA) overnight at 4 °C in a humid chamber. The following day, slices were counterstained with 1:400 dilution of Cell-ID^TM^ Intercalator-Ir (Fluidigm) in DPBS for 30 min at room temperature, washed with DPBS containing 0.1% Triton-X (TC259563, Thermo scientific, Waltham, MA, US) and absolute DPBS twice, and dried at room temperature for at least 20 min. Finally, slices were detected by a Hyperion imaging mass cytometer (Fluidigm) at a resolution of 1 µm and a frequency of 200 Hz. The area of region of interest (ROI) was 1 × 1 mm^2^, and the energy was between 4 and 6. The data were exported as MCD files and .txt files which were read by MCD viewer software. The antibodies panel used in IMC was listed in Appendix A, and all antibodies had positive signals (Appendix A).

MCD files exported from hyperion imaging mass cytometer were exported into tiff files by MCD viewer software. The tiff files were loaded into CellProfiler Version 3.1.9 to generate cell segmentation masks and extract the intensity of markers in the panel [76]. Subsequently, the tiff files with cell masks were imported into histoCAT software Version 1.73 to further analyses [77]. t-SNE analysis was used to transform high dimensional data into two dimensions. The tissue-specific clusters with diverse phenotypes were identified by phonograph analysis based on similar markers expression pattern, and then the clusters were used to run neighborhood analysis to reveal the interaction or avoidance of cell–cell neighbors.

### 4.7. Statistical Analysis

Two-sided Student’s *t* test and Kruskal–Wallis rank sum test were performed for statistical analysis. *p* < 0.05 was considered statistically significant. Spearman rank correlation analysis was used to evaluate the correlation of specific clusters. Overall survival curves of MIBC patients were estimated using Kaplan–Meier analysis and compared by the log-rank test. The software GraphPad Prism 8 was used for statistical analysis.

## 5. Conclusions

In this study, CyTOF and IMC were used to investigate the complexity of the MIBC TME, revealing the heterogeneous characteristic of TME that is more complex than previously understood. We revealed the diverse phenotypes of tumor cells and immune cells of MIBC, and identified a specific cancer stem-like cell cluster that associated with poor prognosis. In addition, our study also dissected the spatial resolution-based phenotypes of immune cells and collagen signatures. Taken together, our study provides a resource of the MIBC TME for future study, and might provide approaches for the precise treatment of MIBC.

## Figures and Tables

**Figure 1 cancers-13-05440-f001:**
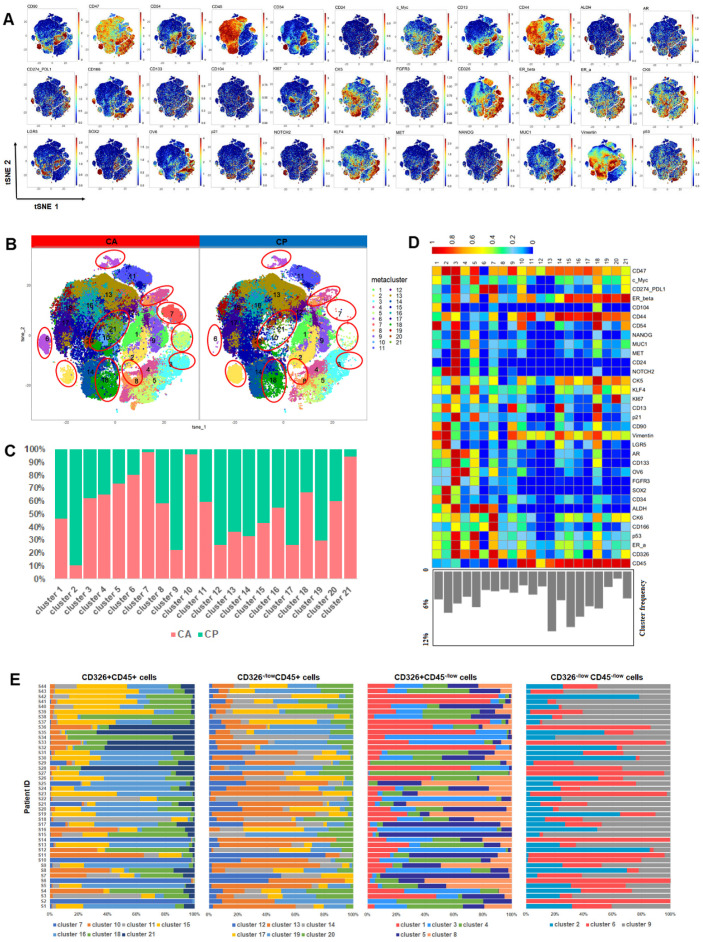
TME landscape in MIBC. (**A**) t-SNE plots of indicated markers of single cells from all samples. (**B**) t-SNE plots of the distinct clusters from cancer (CA) and para-carcinoma (CP) tissues. (**C**) Relative frequencies of 21 clusters in CA and CP tissues. (**D**) Heatmap of indicated markers expression of 21 clusters. (**E**) Relative frequencies of 21 clusters in individual samples.

**Figure 2 cancers-13-05440-f002:**
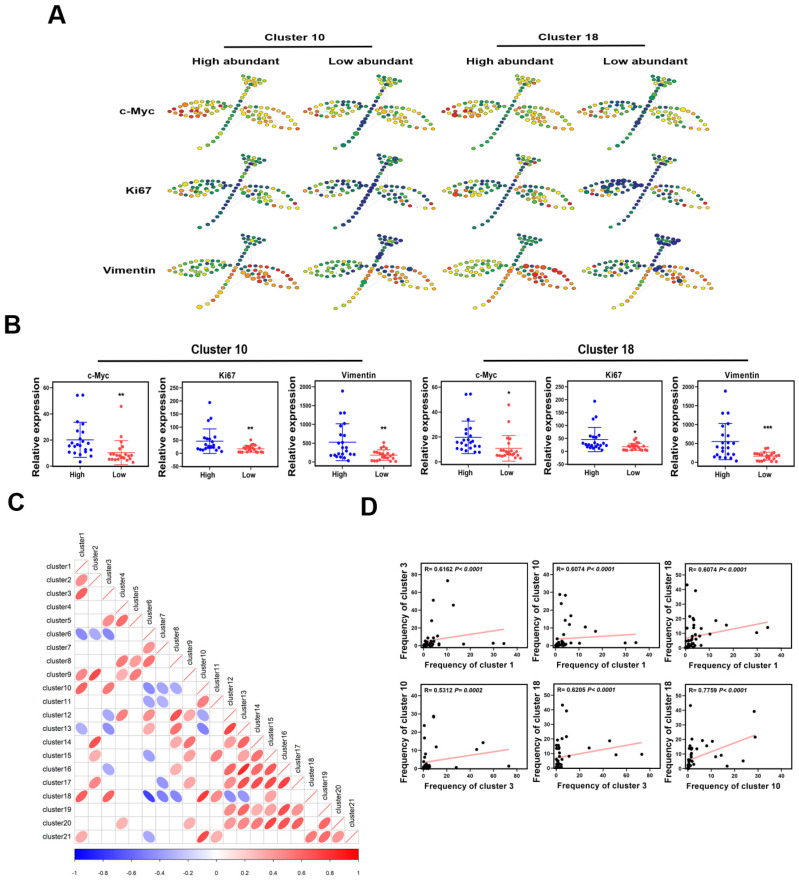
Characteristics of specific clusters. (**A**) SPADE analysis of tumor epithelial cells in cluster 10 or 18 high- and low- abundance groups; the red dots indicate high expression, and the blue dots indicate low expression, the dots size reflect cells numbers. (**B**) c-Myc, Ki67 and vimentin expression levels of tumor epithelial cells in high- and low-abundance group of cluster 10 or 18. (**C**) Spearman rank correlation analysis shows the relationships among clusters in the MIBC TME. (**D**) Spearman rank correlation analysis shows the relationships among cluster 3, cluster 10 and cluster 18.

**Figure 3 cancers-13-05440-f003:**
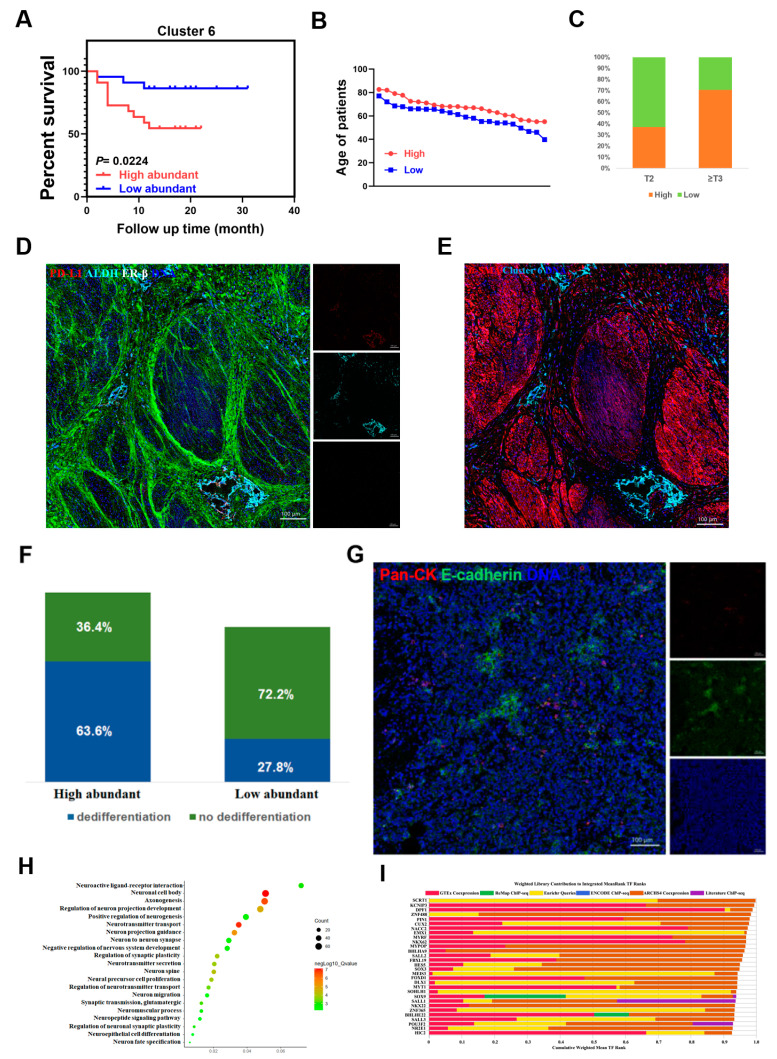
A specific cancer stem-like cell cluster associates with poor prognosis. (**A**) Survival curves show distinctive prognosis of patients in cluster 6 high- and low-abundance group. (**B**) Line plot shows that cluster 6 high-abundance group patients are older than low-abundancy group patients. (**C**) Histogram reveals that the advanced stage patients always belong to cluster 6 high-abundance group. (**D**) Highlighting the position of cluster 6 by ALDH (cyan), PD-L1 (red), ER-β (white), while collagen I presents as green, DNA presents as blue, scale bars are indicated in images. (**E**) IMC analysis reveals that cluster 6 has invaded into muscular layer; cluster 6 presents as cyan, α-SMA presents as red. (**F**) Frequency of dedifferentiation samples in cluster 6 high- and low-abundance groups. (**G**) Images of dedifferentiated tumor cells, Pan-CK presents as red, E-cadherin presents as green, DNA presents as blue. (**H**) Gene ontology (GO) analysis shows that the DEGs of cluster 6 high- and low-abundance groups enrich in neuro-related pathways. (**I**) Transcription factor enrichment analysis reveals the top 30 potential transcription factors of the DEGs between cluster 6 high- and low-abundance groups.

**Figure 4 cancers-13-05440-f004:**
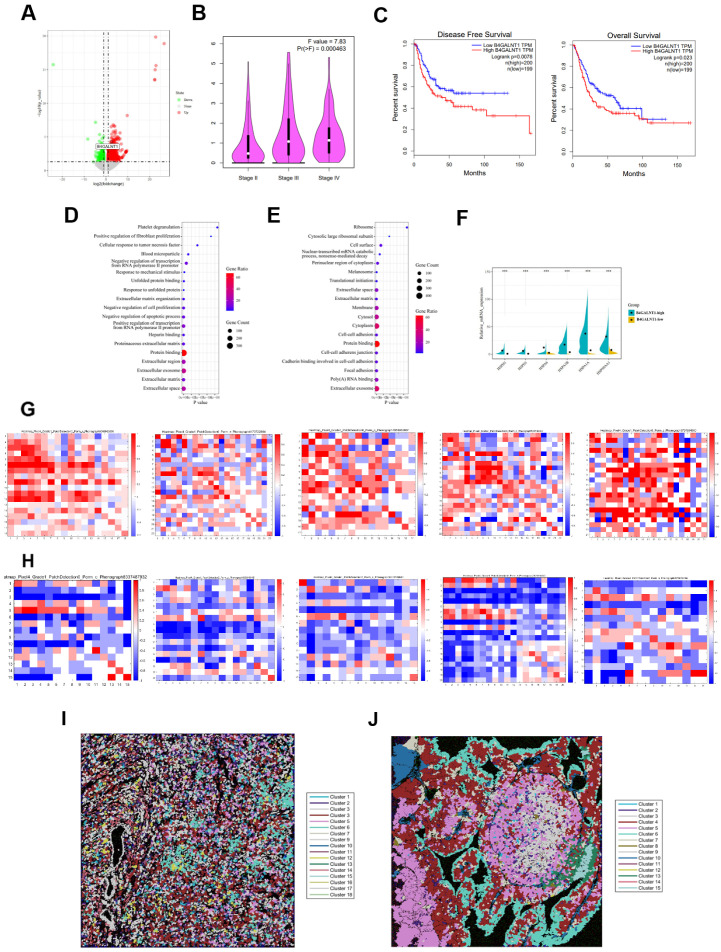
Cluster 6 is potentially regulated by B4GALNT1. (**A**) Volcano plot of the DEGs between cluster 6 high- and low-abundance groups. (**B**) Violin plot shows the B4GALNT1 expression levels of BLCA cohort in TCGA. (**C**) Disease-free survival and overall survival of B4GALNT1 high- and low-expression group patients in TCGA BLCA cohort. (**D**,**E**) GO analysis shows the top 20 pathways in which the DEGs of fibroblasts (**D**) or tumor epithelial cells (**E**) in B4GALNT1 high- and low-expression group are significantly enriched. (**F**) The expression levels of HSPH1, HSPD1, HSPA6, HSPA1B, HSPA1A, HSP90AA1 in B4GALNT1 high- and low-expression groups. (**G**,**H**) Interaction heatmap of 5 B4GALNT1 high-expression patients (**G**) and 5 B4GALNT1 low-expression patients (**H**). The red bands indicate interaction, and the blue bands indicate avoidance. (**I**,**J**) Clusters spatial distribution images of B4GALNT1 high (**I**) and low (**J**) expression patients, which were highlighted by histoCAT software.

**Figure 5 cancers-13-05440-f005:**
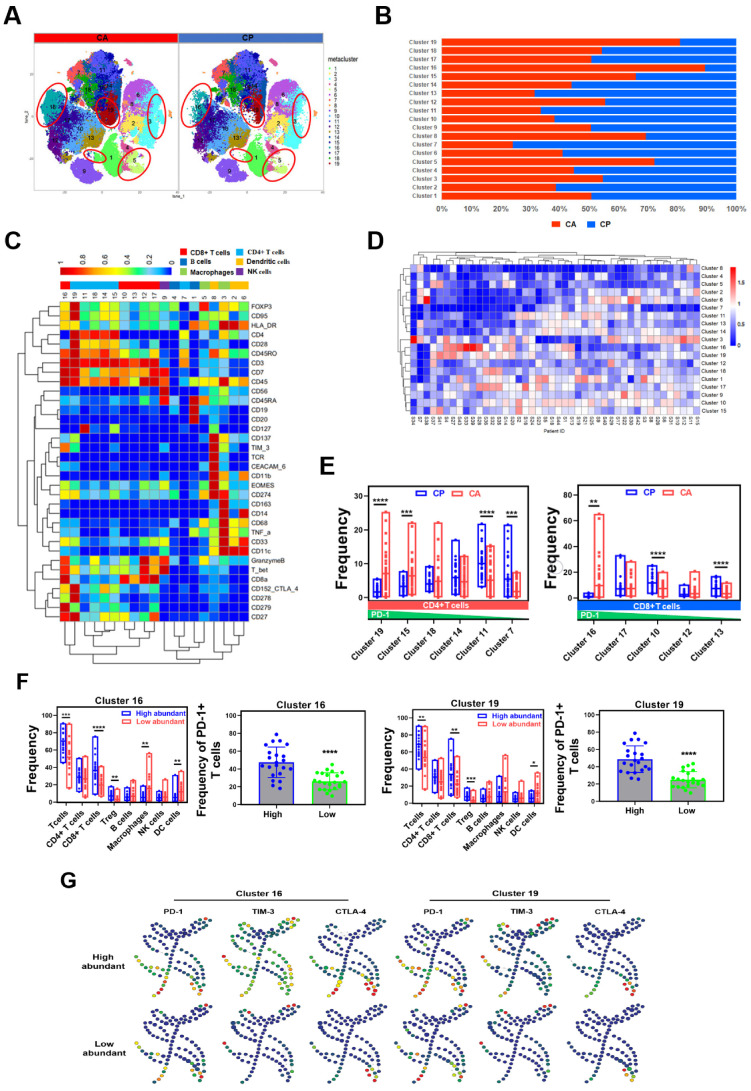
The heterogeneous phenotypes of immune cells in MIBC and the TME. (**A**) t-SNE plot of immune cell clusters from CA and CP tissues. (**B**) Relative frequencies of the 19 immune cell clusters in CA and CP tissues. (**C**) Heatmap of specific markers expression of the 19 immune cell clusters. (**D**) Heatmap of the frequencies of 19 immune cell clusters in each sample. (**E**) Frequencies of specific CD4^+^ T cell clusters and CD8^+^ T cell clusters in CA and CP tissues. (**F**) Frequencies of infiltrating immune cells of high- and low-abundance groups of cluster 16 and 19. (**G**) SPADE analysis of PD-1^+^, TIM-3^+^, CTLA-4^+^ immune cells in cluster 16 and 19 high- and low-abundance groups.

**Figure 6 cancers-13-05440-f006:**
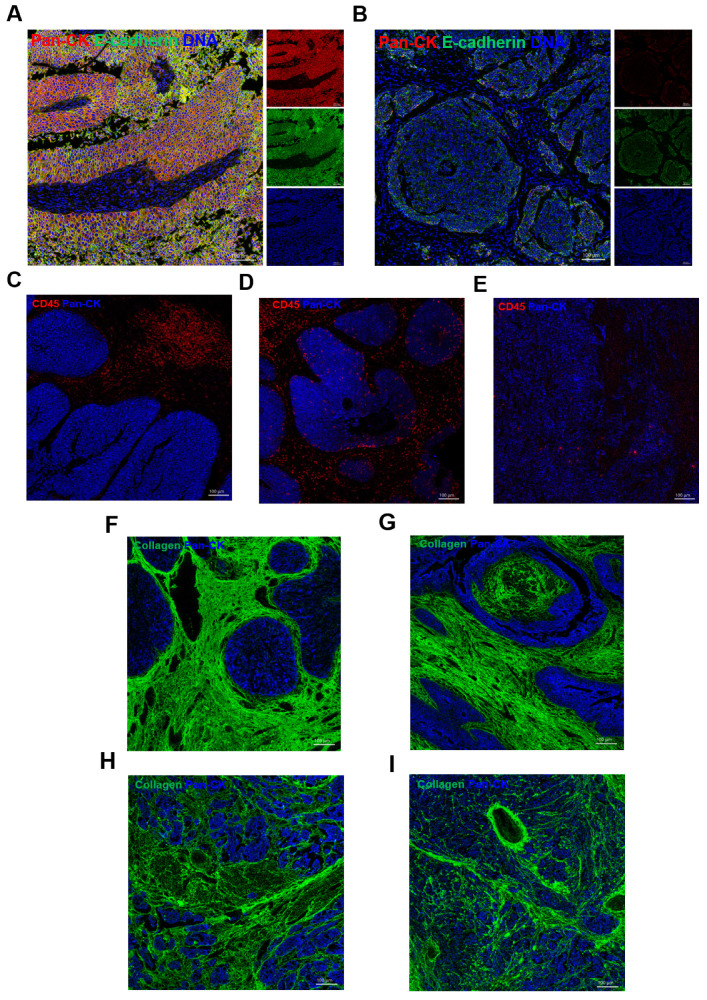
The spatial resolution-based phenotypes of the MIBC TME. (**A**,**B**) Images of differentiated tumor cells (**A**) and dedifferentiated tumor cells (**B**), Pan-CK presents as red, E-cadherin presents as green, DNA presents as blue. (**C**–**E**) Three spatial distribution patterns of immune cells, type I (**C**), type II (**D**), type III (**E**), CD45 presents as red, Pan-CK presents as blue. (**F**–**I**) Four spatial distribution patterns of collagen signatures, type I (**F**), type II (**G**), type III (**H**), type Ⅳ (**I**), Collagen presents as green, Pan-CK presents as blue.

## Data Availability

The data presented in this study are available on request from the corresponding author.

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
