# Peer review of "Single-Cell Proteomic Analysis Dissects the Complexity of Tumor Microenvironment in Muscle Invasive Bladder Cancer"

_cancers, 2021, doi:10.3390/cancers13215440_

Round 1

Reviewer 1 Report

The authors studied the tumor microenvironment of muscle invasive bladder cancer by using mass cytometry and imaging mass cytometry. They identified a specific cancer stem-like cell cluster (ALDH+PD-L1+ER-β-) that was associated with poor prognosis. They also found B4GALNT1 as a potential gene regulating the TME and were correlated with the progression of bladder cancer. However, as the authors mentioned in the limitation part, it is a pity that they didn’t explore the functional analyses of these ALDH+PD-L1+ER-β- cells, such as the ability of invasion or tumorigenicity. This could further prove the stem cell phenotypes in MIBC. Hope this will be done in future publications!

An interesting study with a simple hypothesis and rationale. I do enjoy the analyses performed in this study. Several minor points should be done or clear to spice the manuscript:

  1. Are there any patients receiving PD-1/PD-L1 blockade therapy afterward? Is there any association between the therapeutic responses and the stem cell-like group in the samples?
  2. The authors declared that cluster 6 is related to neural cells. However, these cells are rare in bladder cancer. Do the tumor samples show neural differentiation characteristics in patients with highly expressed cluster 6?
  3. B4GALNT1 is mainly correlated with the fibroblasts in TME. Please describe the role of fibroblasts in bladder cancer TME in the discussion.
  4. Is the patients’ therapeutic outcome in this cohort correlated with the expression level of B4GALNT1?

Author Response

Question 1: Are there any patients receiving PD-1/PD-L1 blockade therapy afterward? Is there any association between the therapeutic responses and the stem cell-like group in the samples?

Response: Thanks for your question and valuable suggestions, due to PD-1/PD-L1 blockade therapy is too expensive and bladder cancer has a low response to it. The PD-1/PD-L1 blockade therapy is not widespread in our area. Therefore, no one in our cohort chose immunotherapy. Due to the small sample size, no association was found between the therapeutic responses and the stem cell-like group patients. However, we plan to sort the cancer stem-like cell cluster (ALDH+PD-L1+ER-β-) from MIBC tissues, and detect the sensitivity of the cancer stem-like cell cluster to drugs by organoid culture. We expect to identified the relationship between the specific therapeutic responses and the cancer stem-like cells.

 Question 2: The authors declared that cluster 6 is related to neural cells. However, these cells are rare in bladder cancer. Do the tumor samples show neural differentiation characteristics in patients with highly expressed cluster 6?

Response: Yes, in patients with highly expressed cluster 6, neural differentiation characteristics can be clearly observed by H&E staining.(the H&E image please find in the attachment  )

 Question 3: B4GALNT1 is mainly correlated with the fibroblasts in TME. Please describe the role of fibroblasts in bladder cancer TME in the discussion.

Response: Thanks for your valuable suggestion. About the role of fibroblasts in bladder cancer TME, previous studies have reported that cancer-associated fibroblasts promote cisplatin resistance, induce EMT and associate with poor prognosis in bladder cancer (PMID: 33033240, 31076571, 30744595). We have added this content in the revised manuscript.

Question 4: Is the patients’ therapeutic outcome in this cohort correlated with the expression level of B4GALNT1?

Response: In this cohort, patients with high-B4GALNT1 expression tended to have worse prognosis compared with the low-B4GALNT1 expression patients, but due to the limited sample size, there was no statistical significance (p=0.0573). Thank you again for your review and suggestions.(the results of survival analysis please find in the attachment  )

Reviewer 2 Report

This is an interesting article to afford many techniques to explore the tumor microenviroment, including the tumor cells and immune cell in single cell status.

But some questions need to be explained.

  1. In the “Results” section, TME landscape in MIBC: Those the cluster1、cluster2、cluster3 and cluster6 were characterized by cancer stem-like cell. Please explain the meaning of cancer stem-like cell. Besides, the author select the cluster6 to be a poor prognosis in the following experiment, please explain it.
  2. In the Figure 1 and Figure 5, the authors use the t-SNE plot tool to distinguish 21 cluster tumor cells and 19 cluster immune cell, respectively. Please explain how to Integrate their correlation due to the focus of article in tumor microenviroment.

Author Response

Question 1: In the “Results” section, TME landscape in MIBC: Those the cluster1cluster2cluster3 and cluster6 were characterized by cancer stem-like cell. Please explain the meaning of cancer stem-like cell. Besides, the author selected the cluster6 to be a poor prognosis in the following experiment, please explain it.

Response: Thank you for your questions. Human cancers harbor a specific population of cancer cells which express stem cell markers such as CD90, LGR5, CD133, and ALDH (PMID: 34445353, 34411919, 30733805), calling cancer stem-like cells. It shows self-renew ability and great cell plasticity (29752993). Previous publications suggested that cancer stem-like cells are the main culprit of cancer relapse, resistance (radiotherapy, hormone therapy, and chemotherapy) and metastasis (PMID: 31369817, 33217316). In this manuscript, we identified 21 clusters in MIBC TME. We stratified patients based on the frequency of each cluster, into high- and low-abundance groups, respectively. Survival analysis showed that among the 21 clusters, only cluster 6 was associated with poor prognosis. Therefore, we selected cluster 6 to further analyzed.

Question 2: In the Figure 1 and Figure 5, the authors use the t-SNE plot tool to distinguish 21 cluster tumor cells and 19 cluster immune cell, respectively. Please explain how to Integrate their correlation due to the focus of article in tumor microenvironment.

Response: In this study, we designed a panel containing 33 TME-related markers to accurately analyze the phenotype of cells in MIBC TME at the single-cell level by CyTOF. By clustering, we detected 21 clusters with diverse phenotypes, including 6 immune cell clusters,19 tumor epithelial cell clusters, and 2 other cell clusters. In this section, we highlighted the phenotypic diversity of tumor epithelial cells and identified a specific cancer stem-like cell cluster, which was associated with poor prognosis. Meanwhile, we found significant heterogeneity in the immune cell clusters which are the key components of TME. Therefore, a panel containing 34 immune-related markers was designed to provide in-depth analysis of the abnormal phenotypes of immune cells, further revealing the complexity of MIBC TME. Thank you again for your review and suggestions.

Reviewer 3 Report

I reviewed the manuscript titled "Single-cell proteomic analysis dissects the complexity of tumor microenvironment in muscle invasive bladder cancer" by Feng and colleagues. The authors used mass cytometry and imaging mass cytometry to analyze tumor cells, immune cells together with TME spatial characteristics of 44 MIBC patients. They report tumor and immune cell clusters with abnormal phenotypes  and a previously unappreciated cancer stem-like cell cluster (ALDH+PD-L1+ER-β-) that was strongly associated with poor prognosis. They also report different spatial architecture characteristics of immune cells (excluded, infiltrated and deserted) and tumor-associated collagens (curved, stretched, directionally distributed and chaotic) in MIBC TME. Overall, the manuscript provides interesting deeper insights on the complexity of MIBC TME at single cell level, that nay help in understanding of heterogeneous characteristics of MIBC. The quality of the figures and their labels need to improve as almost all panels are illegible. The manuscript requires extensive English languish editing.

Author Response

Question 1: The quality of the figures and their labels need to improve as almost all panels are illegible. The manuscript requires extensive English languish editing.

Response: Thanks for your suggestion, we have invited professionals to revise the English of this manuscript. About the illegible Figures, maybe we can provide PPT or PDF version of Figures, it can solve the problem of picture clarity. Thank you again for your review and suggestions.